# Macrolichen Communities Depend on Phorophyte in Conguillío National Park, Chile

**DOI:** 10.3390/plants12132452

**Published:** 2023-06-26

**Authors:** Johana Villagra, Leopoldo G. Sancho, David Alors

**Affiliations:** 1Departamento de Ciencias Agropecuarias y Acuícolas, Facultad de Recursos Naturales, Campus San Juan Pablo II, Universidad Católica de Temuco, Temuco 478 0694, Chile; 2Departamento de Biología y Químicas, Facultad de Recursos Naturales, Campus San Juan Pablo II, Universidad Católica de Temuco, Temuco 478 0694, Chile; dalors@uct.cl; 3Departmento de Farmacología, Farmacognosia y Botánica, Facultad de Farmacia, Universidad Complutense de Madrid (UCM), Madrid 28040, Spain; sancholg@farm.ucm.es

**Keywords:** macrolichens, exposure, *Araucaria araucana*, *Nothofagus antarctica*

## Abstract

The community composition of epiphytic macrolichens from two tree species (*Araucaria araucana* and *Nothofagus antarctica*) was conducted in temperate forests in the Conguillío National Park, Chile. The composition of lichen biota is influenced by phorophyte species, bark pH, and microclimatic conditions. A total of 31 species of macrolichens were found on *A. araucana* and *N. antarctica*. Most of the species showed phorophyte preference, with nine being exclusive to *A. araucana* and 10 to *N. antarctica*. The detrended correspondence analysis (DCA) indicated the formation of three communities: one representing macrolichens growing on *N. antarctica* and two others growing chiefly on *A. araucana*, either with north or south exposure. More work is needed to study the lichen biota of the forests of the Chilean Andes, which are under multiple threats, including clearing and climate change. In order to counteract such risks to native forests and to the biodiversity of the associated epiphytic lichens, conservation plans should be established that consider the factors that influence the composition of the lichen community.

## 1. Introduction

Lichens are symbiotic and poikilohydric organisms [1,2] and are among the most sensitive organisms to environmental change [3,4,5,6,7]. Many studies on epiphytic lichens show a close phorophyte-lichen relationship [8,9,10]. The diversity and distribution of epiphytic lichens respond to different factors, such as the nature of the cortex of the phorophytes [11,12,13], the age of the phorophyte, or the architecture of the tree canopy [14,15,16,17,18,19]. As they contribute to biological diversity and provide important ecosystem services to forests, epiphytic lichens are an important biological group in these ecosystems [20,21,22,23,24].

In addition, the microclimatic variation from the upper part of the canopy to the interior of the forest involves the intensity and direction of light received, the uptake of humidity from the air, and air temperature, which affects their physiology and determines the distribution of epiphytes [25,26,27]. Thus, for example, in the same tree, it is possible to find variable richness and coverage values for epiphytic lichens depending on north-south exposure, as observed by [28] regarding *Nothofagus pumilio* in Isla Navarino (Chile). On the other hand, rain interception by hair lichens growing in the forest canopy could be crucial for the water cycle in rainforests [29]. Anthropic intervention and forest fragmentation internally homogenize forest ecosystems, resulting in less structural heterogeneity [30], with consequences for the diversity of epiphytic lichen communities [31,32]. This may cause local and regional extinctions, including the loss of species that is still unknown to science [33,34,35,36].

In south-central Chile and particularly in the Araucanía region, few studies on the diversity and the structure of lichens communities have been carried out [24,37]. The name of the region stems from the *Araucaria araucana* tree, a relict conifer of the temperate rain forests of South America [38,39,40]. Approximately 75% of the total population of *A. araucana* is found in Chile, mainly in the Araucanía region. Another tree species native to Chile and Argentina is the deciduous *Nothofagus antarctica*, which has the widest ecological range within the genus *Nothofagus* in Chile [41]. *Araucaria araucana* is classified as Endangered [42] and *N. antarctica* as Least Concern [43], in both cases mainly due to anthropogenic disturbances [40,44]. Thus, the present study aimed to characterize the diversity and composition of macrolichens on these two phorophyte species of the native forest of the Andean zone and to assess potential phorophyte preferences and their impact on conservation measures.

## 2. Results

### 2.1. Diversity of Macrolichens

A total of 31 species of macrolichens were found for both phorophyte species (Table 1). The families with the highest number of species recorded were Parmeliaceae, with 16 species (52%), and Peltigeraceae (subfamily Lobarioideae), with seven species (23%). The genera *Pseudocyphellaria*, *Parmelia*, and *Protousnea* had five, four, and three species, respectively. We found 21 species of macrolichens on *Araucaria araucana* and 22 on *Nothofagus antarctica*. According to the Shannon-Wiener diversity index (H′), *A. araucana* showed higher lichen diversity values than *N. antarctica* (Mann–Whitney test: U = 2107; z-score = 2.649; *p* = 0.008). We did not find significant differences relative to the vertical distribution of the subsamples (*A. araucana*: F = 0.09; gl = 2; *p* = 0.917; *N. antarctica*: F = 0.12; gl = 2; *p* = 0.887).

Foliose lichens were dominant in both tree species in terms of the number of species (62% in *A. araucana*; 60% in *N. antarctica*). The main photobiont type was green algae on both tree species, representing 74% of the species. Specifically, *A. araucana* photobiont green-type algae represent 86% of the species and 64% of the species on *N. antarctica*. Regarding the inferred geographical distribution of the identified lichens, both phorophytes endemic species have a higher representation than other distribution types, with 38.1% in *A. araucana* and 36.4% in *N. antarctica* (Table 1).

The coverage percentage of the macrolichens studied did not show differences with respect to the phorophyte (t = 0.00179; *p* = 0.998). *Platismatia glauca* and *Protousnea poeppigii* presented higher coverage (14.1–19.2%) on *A. araucana*. *Pseudocyphellaria coriifolia* (38.6%) and *Pseudocyphellaria citrina* (12.7%) presented higher coverage on *N. antarctica*. Nine and 10 species were exclusive to *A. araucana* and *N. antarctica*, respectively, and 12 species were shared by the two phorophyte species (Table 2).

### 2.2. Assembly of Species

The DCA (Figure 1) showed three groups of species: one group, including *Pseudocyphellaria coriifolia*, *P. citrina*, *Nephroma cellulosum*, and *Podostictina scabrosa*, with higher coverage for *Nothofagus antarctica* (group 1 in Figure 1), and two groups of species with higher coverage for *Araucaria araucana*: one with higher coverage for north exposure, like *Protousnea poeppigii*, *Platismatia glauca*, and *Coelopogon epiphorellus* (group 2 in Figure 1), and the other with higher coverage for south exposure, like *Nephroma antarcticum*, *Podostictina flavicans*, *Parmelia saxatilis*, and *Pseudocyphellaria granulata* (group 3 in Figure 1).

### 2.3. Change in Lichens Community in North vs. South Exposure on Araucaria araucana

The south (S) exposure on *A. araucana* showed a higher number of species (17) reaching a higher coverage percentage (75%), and the dominant biotype was foliose, while for north (N) exposure, the number of species (14) and coverage percentage (59%) was lower, and the dominant biotype was fruticose. Ten species were present in both exposures (Table 2). Some of these showed higher coverage percentage for S exposure: *Nephroma antarcticum* (Mann–Whitney test: U = 398; z-score = 3.626; *p* < 0.001), *Parmelia saxatilis* (U = 403; z-score = 3.65; *p* < 0.001), *Platismatia glauca* (U = 409; z-score = 3.00; *p* < 0.001), and *Pseudocyphellaria granulata* (U = 514; z-score = 2.27; *p* = 0.023). On the other hand, *Protousnea poeppigii* (U = 211; z-score = 5.313; *p* < 0.001) and *Coelopogon epiphorellus* (U = 175; z-score = 5.742; *p* < 0.001) presented a higher coverage percentage for N exposure.

### 2.4. Microclimate

The microclimatic parameters (mean temperature and humidity) at the different sites during the research period showed significant differences in the southern and northern exposures of *A. araucana* (*p* < 0.05; Table 3).

The higher differences in average temperature and monthly relative humidity are detailed in Figure 2. Significant differences were observed mainly in the southern, autumn-winter months and also in February for the average temperature. The maximal temperatures (Tmax) were significantly different, and no significant differences were found between the minimal temperatures (Tmin), maximal H (%), and minimal humidity.

### 2.5. Bark pH

The mean value and statistical error of bark pH were 4.62 ± 0.03 in *A. araucana* and 5.79 ± 0.04 in *N. antarctica*. These results showed statistical differences between the two tree species (*p* < 0.001; t = 21.821) and more acidic bark in *A. araucana* than in *N. antarctica*.

## 3. Discussion

This study characterized three communities of macrolichens in a mixed forest of *Araucaria araucana* and *N. antarctica*. More than a half of the species were exclusive to one of the phorophytes species; however, the richness of the species was similar between both phorophytes. The sampling was restricted to tree trunks; therefore, the number of species may be an underestimate and the percentages of exclusive/shared species may have differed if the tree branches were sampled as well.

The lichen community found on *Nothofagus antarctica* (Group 1) is characterized by a higher abundance of cyanolichens, such as *Pseudocyphellaria coriifolia*, *Pseudocyphellaria citrina*, *Nephroma cellulosum*, *Podostictina scabrosa*, coinciding with previous works that also suggest that deciduous trees are favorable for the establishment of cyanolichens [46,47,48]. The majority of cyanolichens are usually found in environments with low light intensities, high humidity, and requiring water in liquid form to carry out photosynthesis [49,50]. According to Kussinen [51] and Hedenâs and Ericson [52], the cyanolichens can be used as indicators of habitat stability; therefore, it could be that the higher frequency of the cyanolichens is an indicator of better habitat conservation and forest health.

We found a higher degree of richness for the macrolichen species in the south-facing trunks, showing that the higher temperatures and humidity from a previous study in a *Nothofagus pumilio* forest on Navarino Island (the Magellan and Chilean Antarctic Region) [28] coincide with our findings. The lichen communities found on *A. araucana* are dominated by the fruticose species *Protousnea poeppigii* for N exposure and by foliose species *Nephroma antarcticum* and *Podostictina flavicans* for S exposure. The differences between the S and N exposures in *A. araucana* was attributed to the microclimatic variables evaluated (temperature and humidity), which coincide with previous studies that have shown how these variables determine the structure of lichen communities [15,17,53].

On the other hand, the N exposure of *A. araucana* showed more fruticose lichens, a result which is in accordance with Woda et al. [54], who found abundant fruticose lichens in young Fitzroyetum forests in the temperate rainforests of southern Chile’s coastal range Cordillera Pelada, suggesting that fruticose lichens may prefer microhabitats with greater exposure to light, and also showing a higher abundance in stands with higher luminosity. Other abiotic conditions are important to lichens in cold and mountainous habitats, such as hoarfrost and snow, which can negatively affect lichens [55] more in the south than in the north, which receives more sunlight and reaches higher temperatures (Figure 2). These factors affect fruiting lichens more; because of their shape, they accumulate more frost and snow and can become detached or break.

The differences showed in the DCA analysis are primarily attributable to the phorophyte species and, secondly, to exposure. The differences in lichen species composition, depending on phorophyte and phorophyte preferences, were also reported in other studies with *Populus tremula* in Sweden [52], oak forests in Costa Rica [56], alpine spruce forests [57], and also one work in which the different tree species in Conguillío National Park were analyzed [24]. We can infer that some species, such as *Protousnea fibrillata*, *Protousnea magellanica*, *Podostictina flavicans*, *Pseudocyphellaria faveolata*, or *P. glabra*, prefer rough bark and the higher radiation typical of *A. araucana* forests, while other species, such as *Collema glaucophthalmum*, *Pannaria farinosa*, *Pseudocyphellaria hirsuta*, or *Podostictina scabrosa*, have a higher affinity with less compact bark and greater protection through the canopy of *N. antarctica*. This suggests that the microenvironment is associated with bark trees, as the physical and chemical properties of the phorophyte surface may determine the composition of lichen species on *A. araucana* and *N. antarctica*. One of these factors could be the pH of the tree bark, which we measured as being different between *A. araucana* and *N. antarctica* by more than one pH point, and this has been found to be a determinant in studies in Atlantic Brazilian rainforests, premontane Colombian forests [9], and even in the *Nothofagus* trees from Central Chile [13]. Both phorophyte species studied here have rough bark, but the bark of *A. araucana* is much thicker, reaching up to 20 cm [40], showing higher grooves between the bark plaques and allowing for specialized microenvironments.

This work represents a contribution in terms of a contribution to mitigating the scarcity of knowledge on lichens and other epiphytes, which are of great importance to the conservation of biodiversity in Chile [58,59,60,61,62,63]. It is necessary to better understand the hidden diversity of epiphytic lichens to assess the real diversity of the forests of Chile, as well as to have well-founded data on the structure and function of these communities. Therefore, it is necessary to carry out more integrated studies of the diversity of epiphytic lichens in order to conserve the maximum diversity of species through the development of management plans in natural forested areas.

## 4. Materials and Methods

### 4.1. Study Area

The lichen biota was studied in a mixed stand of *Araucaria araucana* (Mol.) K. Koch. and *Nothofagus antarctica* (G. Forst.) Oerst. in the Conguillío National Park (38°39′05.62″ S and 71°38′51.68″ W), located in the pre-mountain range of Los Andes in the province of Cautín, Araucanía Region, Chile (Figure 3). *Araucaria araucana* is a native conifer that is native to Chile and Argentina, with a very restricted distribution [40,64] and its range being affected by climate change in the Araucanía Region [65]. The climate of the study area is temperate-cold, with marked contrasts between prolonged winters with low temperatures and dry summers with high temperatures. The average annual temperature is 8.6 °C, with an average of 15.1 °C in January (warmest month) and 1.9 °C in July (coldest month) [66]. The average annual precipitation ranges from 2500 to 3000 mm [67], even though much of the precipitation is in the form of snow (from May to September).

### 4.2. Sampling Methodology

Five 30 × 30 m plots, each 150 m apart, were located in the stands of *A. araucana*-*N. antarctica.* Within each plot, we selected five old trees per phorophyte species, for a total of 50 trees. For *A. araucana,* we sampled trees with a trunk perimeter no lower than 180 cm, and for *N. antarctica,* we sampled trees with a perimeter of at least 40 cm for the trunk diameter. The richness, coverage, and vertical distribution of the lichen communities were evaluated by applying 30 × 20 cm quadrats directly to the trunk at 20, 80, and 150 cm heights, respectively, thus obtaining 75 subsamples for each phorophyte species. A total of 38 subsamples were conducted on the S side and 37 subsamples on the N side of the trees. We evaluated the assemblage of macrolichens present in the two different phorophyte species. 

### 4.3. Taxonomic Determination

The identification of the lichen species was conducted using specific literature, including Galloway [68,69,70], White and James [71], Stenroos [72], Wedin [73], Bjerke et al. [74,75], Calvelo et al. [76], Boluda et al. [33], and Passo et al. [77]. The material was observed under a stereomicroscope (Leica Wild M8) and a microscope (Nikon Eclipse 80i). The identification of the species included chemical tests. The reagents used were K (10% saturated KOH solution), C (5% aqueous sodium hypochlorite solution), and KC (application of K followed by C). The vouchers of representative individuals were deposited in the MAF-Lich herbarium of the Faculty of Pharmacy in Madrid (number: MAF25054-MAF25103).

### 4.4. Biogeographic Distribution Categories

Biogeographic distribution categories are indicated according to a simplification of [45]. We used five categories: Endemic (present only in southern South America); Austral (species with Paleoaustral distribution: lichens are thought to represent primitive Gondwanan groups. These would date from the Cretaceous or earlier and Neoaustral distribution: lichens are taxa dispersed after the fragmentation of Gondwanaland, mainly between post-Oligocene and the present); Cosmopolitan (species with worldwide distribution); Tropical (includes Neotropical species and tropical species with a wider distribution); South American-African.

### 4.5. Microclimate Evaluation

Microclimatic data were obtained for two different sites and were recorded for 2 h each over 333 days from 25 April 2022 to 24 March 2023 using four data loggers (iButton^®^ temperature logger model DS1922). Data Loggers were placed directly on the trunk of *Araucaria araucana* (n = 2). Each of them was placed on the southern and northern exposure of this tree species.

### 4.6. Measurement of Bark pH

Five trees of each species (*A. araucana* and *N. antarctica*) were sampled, and three replicates were taken from each tree. The 10 × 10 cm bark samples were air-dried and stored in paper bags until analysis in the laboratory. For the determination of bark surface pH, the methodology of [13], which is based on that of [78], was used. For each sample, we crushed 2 g of bark and left it to soak in 30 mL of distilled water, adjusting the pH to 7.0. After 24 h, the pH was measured with an Elmetron CX 701 pH meter.

### 4.7. Statistical Analyses 

For richness, coverage, and vertical distribution, a community diversity analysis of the lichen was undertaken using the program PAST 4,11. For each subsample studied, species richness was registered, and the Shannon–Wiener diversity index (H′) was calculated according to the formula:$${{}^{\circ}H}^{\prime }=-\sum_{i=1}^{n}=\mathrm{piln}(\mathrm{pi})$$
where pi = relative proportion (coverage) of the i species. 

A detrended correspondence analysis (DCA) was performed in order to evaluate the influence of the habitats on the distribution of the species according to their abundance. This analysis allows for spatially ordering species according to the habitats studied, excluding species with coverage below 1%.

The vertical distribution of the lichen species for each phorophyte species (based on relative abundance per subsample) was analyzed using a two-way ANOVA. 

## Figures and Tables

**Figure 1 plants-12-02452-f001:**
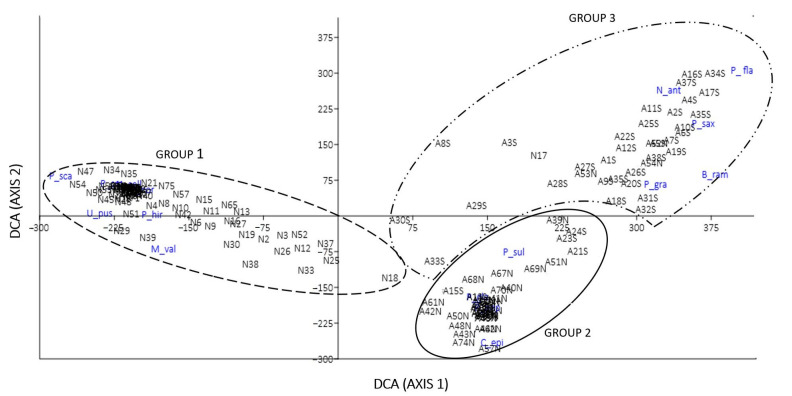
Detrended correspondence analysis (DCA) figure. Lichen surveys are plotted on two axes forming three groups: group 1: species from *Nothofagus antarctica;* group 2: species for north exposure on *A. araucana*; group 3: species for south exposure on *A. araucaria*. Text in blue are the species names.

**Figure 2 plants-12-02452-f002:**
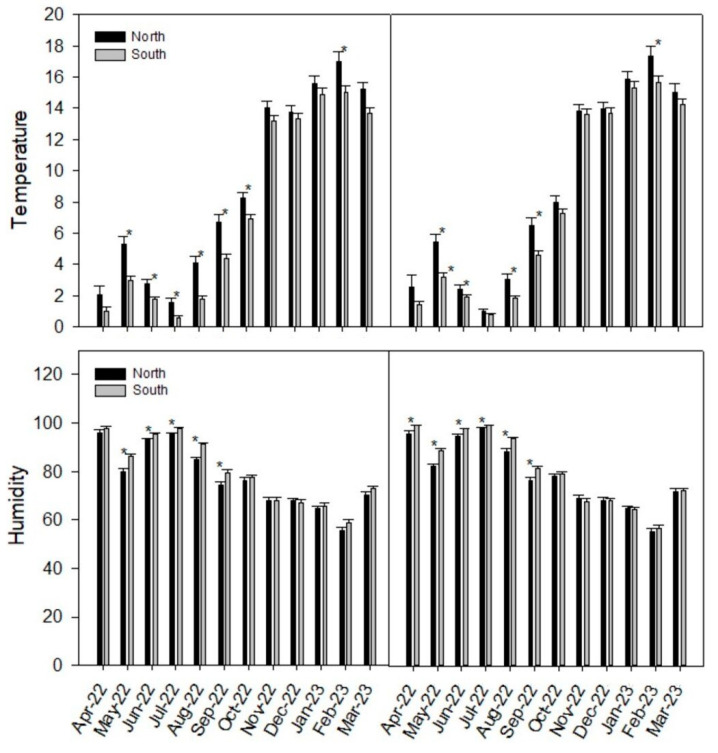
Monthly temperature and humidity values. Temperature (**upper case**) and relative humidity (**lower case**) were measured at two sites (**left** and **right panels**) with four data loggers (n = 2). Significant differences are marked by an asterisk.

**Figure 3 plants-12-02452-f003:**
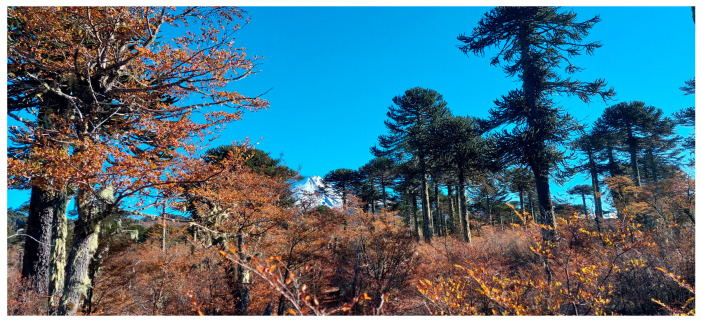
Mixed forest of *A. araucana* (righ upper side) and *N. antarctica* (left and botton side) at Coguillío National Park. In the image we can appreciate snow at summit of the Llaima.

**Table 1 plants-12-02452-t001:** List of lichens registered in *Araucaria araucana* and *Nothofagus antarctica* in Conguillío National Park, Chile. The table shows growth form, family, main photobiont type, and geographical distribution according to the bibliography of the species sampled in this work.

Species Name	Growth Form	Family	Photobiont	Geographical Distribution *
*Bryoria araucana* Boluda, D. Hawksw. & V.J. Rico	Fruticose	Parmeliaceae	A	Endemic
*Bunodophoron ramuliferum* (I.M. Lamb) Wedin	Fruticose	Sphaerophoraceae	A	Austral
*Cladonia* sp.	Dimorphic	Cladoniaceae	A	Not applicable
*Coelopogon epiphorellus* (Nyl.) Brusse & Kärnefelt	Fruticose	Parmeliaceae	A	South American-African taxa
*Collema glaucophthalmum* Nyl.	Gelatinous	Collemataceae	C	Pantropical
*Hypotrachyna sinuosa* (Sm.) Hale	Foliose	Parmeliaceae	A	Cosmopolitan
*Menegazzia valdiviensis* (Räsänen) R. Sant.	Foliose	Parmeliaceae	A	Endemic
*Nephroma antarcticum* (Wulfen) Nyl.	Foliose	Nephromataceae	A	Endemic
*Nephroma cellulosum* (Ach.) Ach.	Foliose	Nephromataceae	C	Austral
*Notoparmelia protosulcata* (Hale) A. Crespo, Ferencova & Divakar	Foliose	Parmeliaceae	A	Austral
*Pannaria farinosa* Elvebakk & Fritt-Rasm	Squamulose	Pannariaceae	A	Austral
*Pannoparmelia angustata* (Pers.) Zahlbr.	Foliose	Parmeliaceae	A	Austral
*Pannoparmelia wilsonii* (Räsänen) D.J. Galloway	Foliose	Parmeliaceae	A	Endemic
*Parmelia saxatilis* (L.) Ach.	Foliose	Parmeliaceae	A	Cosmopolitan
*Parmelia* sp.	Foliose	Parmeliaceae	A	Not applicable
*Parmelia sulcata* Taylor.	Foliose	Parmeliaceae	A	Cosmopolitan
*Peltigera polydactylon* (Neck.) Hoffm.	Foliose	Peltigeraceae	C	Cosmopolitan
*Platismatia glauca* (L.) W.L. Culb. & C.F. Culb.	Foliose	Parmeliaceae	A	Cosmopolitan
*Protousnea fibrillata* Calvelo, Stock.-Wörg., Liber. & Elix	Fruticose	Parmeliaceae	A	Endemic
*Protousnea magellanica* (Mont.) Krog	Fruticose	Parmeliaceae	A	Endemic
*Protousnea poeppigii* (Nees & Flot.) Krog	Fruticose	Parmeliaceae	A	Endemic
*Podostictina flavicans* (Hook. f. & Taylor) Moncada & Lücking	Foliose	Lobariaceae	A	Endemic
*Podostictina scabrosa* (R. Sant.) D.J. Galloway & de Lange	Foliose	Lobariaceae	C	Endemic
*Pseudocyphellaria coriifolia* (Müll.Arg.) Malme	Foliose	Lobariaceae	C	Endemic
*Pseudocyphellaria citrina* (Gyeln.) Lücking, Moncada & S. Stenroos	Foliose	Lobariaceae	C	Bipolar
*Pseudocyphellaria granulata* (C. Bab.) Malme	Foliose	Lobariaceae	A	Austral
*Pseudocyphellaria faveolata* (Delise) Malme	Foliose	Lobariaceae	A	Austral
*Pseudocyphellaria hirsuta* (Mont.) Malme	Foliose	Lobariaceae	C	Endemic
*Psoroma hypnorum var. hypnorum* (Vahl) Gray	Squamulose	Pannariaceae	C	Bipolar
*Usnea pusilla* (Rasanen)	Fruticose	Parmeliaceae	A	Austral
*Usnea* sp.	Fruticose	Parmeliaceae	A	Not applicable
*Bryoria araucana* Boluda, D. Hawksw. & V.J. Rico	Fruticose	Parmeliaceae	A	Endemic
*Bunodophoron ramuliferum* (I.M. Lamb) Wedin	Fruticose	Sphaerophoraceae	A	Austral
*Cladonia* sp.	Dimorphic	Cladoniaceae	A	Not applicable
*Coelopogon epiphorellus* (Nyl.) Brusse & Kärnefelt	Fruticose	Parmeliaceae	A	South American-African taxa
*Collema glaucophthalmum* Nyl.	Gelatinous	Collemataceae	C	Pantropical
*Hypotrachyna sinuosa* (Sm.) Hale	Foliose	Parmeliaceae	A	Cosmopolitan
*Menegazzia valdiviensis* (Räsänen) R. Sant.	Foliose	Parmeliaceae	A	Endemic
*Nephroma antarcticum* (Wulfen) Nyl.	Foliose	Nephromataceae	A	Endemic
*Nephroma cellulosum* (Ach.) Ach.	Foliose	Nephromataceae	C	Austral
*Notoparmelia protosulcata* (Hale) A. Crespo, Ferencova & Divakar	Foliose	Parmeliaceae	A	Austral
*Pannaria farinosa* Elvebakk & Fritt-Rasm	Squamulose	Pannariaceae	A	Austral
*Pannoparmelia angustata* (Pers.) Zahlbr.	Foliose	Parmeliaceae	A	Austral
*Pannoparmelia wilsonii* (Räsänen) D.J. Galloway	Foliose	Parmeliaceae	A	Endemic
*Parmelia saxatilis* (L.) Ach.	Foliose	Parmeliaceae	A	Cosmopolitan
*Parmelia* sp.	Foliose	Parmeliaceae	A	Not applicable
*Parmelia sulcata* Taylor.	Foliose	Parmeliaceae	A	Cosmopolitan
*Peltigera polydactylon* (Neck.) Hoffm.	Foliose	Peltigeraceae	C	Cosmopolitan
*Platismatia glauca* (L.) W.L. Culb. & C.F. Culb.	Foliose	Parmeliaceae	A	Cosmopolitan
*Protousnea fibrillata* Calvelo, Stock.-Wörg., Liber. & Elix	Fruticose	Parmeliaceae	A	Endemic
*Protousnea magellanica* (Mont.) Krog	Fruticose	Parmeliaceae	A	Endemic
*Protousnea poeppigii* (Nees & Flot.) Krog	Fruticose	Parmeliaceae	A	Endemic
*Podostictina flavicans* (Hook. f. & Taylor) Moncada & Lücking	Foliose	Lobariaceae	A	Endemic
*Podostictina scabrosa* (R. Sant.) D.J. Galloway & de Lange	Foliose	Lobariaceae	C	Endemic
*Pseudocyphellaria coriifolia* (Müll.Arg.) Malme	Foliose	Lobariaceae	C	Endemic
*Pseudocyphellaria citrina* (Gyeln.) Lücking, Moncada & S. Stenroos	Foliose	Lobariaceae	C	Bipolar
*Pseudocyphellaria granulata* (C. Bab.) Malme	Foliose	Lobariaceae	A	Austral
*Pseudocyphellaria faveolata* (Delise) Malme	Foliose	Lobariaceae	A	Austral
*Pseudocyphellaria hirsuta* (Mont.) Malme	Foliose	Lobariaceae	C	Endemic
*Psoroma hypnorum var. hypnorum* (Vahl) Gray	Squamulose	Pannariaceae	C	Bipolar
*Usnea pusilla* (Rasanen)	Fruticose	Parmeliaceae	A	Austral
*Usnea* sp.	Fruticose	Parmeliaceae	A	Not applicable

* Austral (species with Paleoaustral and Neoaustral distribution). Cosmopolitan (species with worldwide distribution); endemic (present only in southern South America); tropical (includes Neotropical species and tropical species with a wider distribution) according to Galloway [45].

**Table 2 plants-12-02452-t002:** Mean values (±standard error) of coverage percentage data of the macrolichens on the two studied phorophytes and for the two exposure conditions (separately) in *A. araucana*. Asterisks (*) indicate significant differences between the percentages of the nine shared species (n = 75).

Species	*N. antarctica*	*A. araucana*	*A. araucana* South	*A. araucana* North
*P. poeppigii*		19.2 ± 1.67	5.18 ± 1.0 *	34.10 ± 2.5
*P. glauca*	5.72 ± 1.47	16.7 ± 1.29	9.39 ± 1.3 *	24.50 ± 2.1
*C. epiphorellus*	4.52 ± 1.30	12.4 ± 1.25	3.17 ± 0.9 *	23.60 ± 1.88
*N. antarcticum*	2.04 ± 0.61	11.2 ± 1.57	16.71 ± 2.5 *	5.60 ± 1.73
*P. sulcata*	1.13 ± 0.95	2.84 ± 0.45	1.93 ± 0.5	4.00 ± 0.76
*B. araucana*	0.64 ± 0.27	0.58 ± 0.17	0.90 ± 0.3	0.30 ± 0.19
*N. cellulosum*	0.37 ± 0.30	0.18 ± 0.08	0.20 ± 0.1	0.20 ± 0.4
*P. coriifolia*	9.70 ± 1.71	1.93 ± 0.63	3.50 ± 1.1	-
*P. citrina*	38.57 ± 2.74	0.42 ± 0.16	0.80 ± 0.3	-
*P. wilsonii*	12.74 ± 2.53	0.34 ± 0.14	-	0.80 ± 0.28
*U. pusilla*	0.18 ± 0.14	0.16 ± 0.01	-	0.40 ± 0.2
*Usnea sp.*	6.90 ± 1.24	0.10 ± 0.06	-	0.20 ± 0.1
*P. saxatilis*	0.35 ± 0.17	7.73 ± 1.22	12.99 ± 2.2 *	2.00 ± 0.64
*P. granulata*	-	5.05 ± 0.71	6.30 ± 1.0 *	3.90 ± 1.0
*P. flavicans*	-	10.10 ± 1.52	18.20 ± 2.7	-
*P. magellanica*	-	4.68 ± 1.17	8.89 ± 2.3	-
*B. ramuliferum*	-	3.58 ± 0.88	6.40 ± 1.7	-
*P. faveolata*	-	2.05 ± 0.58	3.70 ± 1.1	-
*P. fibrillata*	-	0.42 ± 0.18	0.80 ± 0.3	-
*Parmelia sp.*	-	0.12 ± 0.01	0.20 ± 0.1	-
*N. protosulcata*	-	0.24 ± 0.16	-	0.50 ± 0.32
*P. scabrosa*	-	-	-	-
*M. valdiviensis*	7.19 ± 1.58	-	-	-
*P. hirsuta*	3.06 ± 0.88	-	-	-
*P. farinosa*	2.87 ± 0.73	-	-	-
*P. polydactylon*	1.86 ± 0.73	-	-	-
*Cladonia sp.*	0.57 ± 0.48	-	-	-
*C. glaucophthalmum*	0.40 ± 0.33	-	-	-
*P. angustata*	0.32 ± 0.19	-	-	-
*P. hypnorum*	0.32 ± 0.18	-	-	-
*H. sinuosa*	0.30 ± 0.25	-	-	-

**Table 3 plants-12-02452-t003:** Annual mean temperature and relative humidity from microclimatic sensors.

	T North (Site A)	T South (Site A)	T North (Site B)	T South (Site B)	H North (Site A)	H South (Site A)	H North (Site B)	H South (Site B)
Media	7.75 ± 0.13	8.07 ± 0.129	9.18 ± 0.153	9 ± 0.16	78.74 ± 0.38	79.64 ± 0.36	75.99 ± 0.41	77.59 ± 0.41
Máximum	36.542		37.526		52.039		48.581	
Minimum	−10.043		−9.543		−10.005		−10.484	
t	7.075		4.482		1.700		3.732	
*p*	>0.05		>0.05		0.089		0.00019	

T: temperature in °C, H: relative humidity (%), t: student’s *t*-test, and *p*: *p*-value.

## Data Availability

The vouchers of lichen samples were deposited in MAF-Lich herbarium of the Faculty of Pharmacy in Madrid (numbers MAF25054-MAF25103).

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
