# Peer review of "Macrolichen Communities Depend on Phorophyte in Conguillío National Park, Chile"

_plants, 2023, doi:10.3390/plants12132452_

Round 1
Reviewer 1 Report
Results macro lichen richness rather than richness lichens would read better.
line 175-177 associated (to) with would be a better description of the correlation.
hoar frost formation might be a critical factor in the distribution of fruticose lichen on tree trunks. A discussion about this and the prevailing wind might be appropriate?
Fruticose lichens are abundant on Nothofagus trees but tend to occur on lateral branches which were not well sampled in this study's methods. A discussion of how the sampling method influenced the taxa captured.
The lat and long for this study area may be correct but, for me I could not find it with my mapping program. Check that it is correct. It also maybe best to report in degrees decimal. Or just include a map that also outlines the distribution of the Araucaria forest types.
More background information on why Araucaria forests are burned and becoming less common might be appropriate in the Introduction.
In the discussion more detail could be provide for the reader on the density, texture and roughness of the tree's bark could be provided. Araucaria bark texture is so different as it ages.
Canopy structure of the two trees are very different and a diagram, photo or description of the canopy shapes would add to the readers understanding of the results. Especially for readers that are not that familiar with the tree species.
Line 175 --
For correlation of results it would be better to use the word "with" rather than to.
The English is good.
Author Response
Thank you so much for your vaulable comments, find our responses in the attached word file.

Reviewer 2 Report
The present paper is a well written paper on phorophyte preference for lichens in Chile.
The pH of the bark should have been measured. Bark pH may often be important for which species are found as you have shortly mentioned in the discussion. Are any data on differences in pH between Arucaria and Nothofagus available?
It would be good to describe the size of the trees (appoximate height and trunk diameter) and density of the stands.
Light conditions should have been described in some way . Are there any differences in light levels for the 30x20 cm quadrats used for sampling between Arucaria and Nothofagus?
With more data on e.g. light conditions and bark pH it would have been easier to discuss why the different lichen species have different preferences for the two phorophytes.
The same temperature and humidity data are presented twice in both Figure 2 and Table 4. I suggest to delete Table 4.
Please check that genera and species names are written with italic font throughout the paper.
In Table 3 there is some Spanish (?). Máximo should be maximum and Mínimo should be minimum.
Author Response

(The authors gave the same response as above.)
